# Microfabricated electrodes unravel the role of interfaces in multicomponent copper-based $CO_2$ reduction catalysts

Gastón O. Larrazábal [1], Tatsuya Shinagawa [1], Antonio J. Martín [1] & Javier Pérez-Ramírez [1]

The emergence of synergistic effects in multicomponent catalysts can result in breakthrough advances in the electrochemical reduction of carbon dioxide. Copper-indium catalysts show high performance toward carbon monoxide production but also extensive structural and compositional changes under operation. The origin of the synergistic effect and the nature of the active phase are not well understood, thus hindering optimization efforts. Here we develop a platform that sheds light into these aspects, based on microfabricated model electrodes that are evaluated under conventional experimental conditions. The relationship among the electrode performance, geometry and composition associates the high carbon monoxide evolution activity of copper-indium catalysts to indium-poor bimetallic phases, which are formed upon exposure to reaction conditions in the vicinity of the interfaces between copper oxide and an indium source. The exploratory extension of this approach to the copper-tin system demonstrates its versatility and potential for the study of complex multicomponent electrocatalysts.

[1] Institute for Chemical and Bioengineering, Department of Chemistry and Applied Biosciences, ETH Zurich, Vladimir-Prelog-Weg 1, Zurich 8093, Switzerland. Correspondence and requests for materials should be addressed to J.P.-R. (email: jpr@chem.ethz.ch)

C ombining the electrochemical reduction of $CO_2$ (e$CO_2$RR) with carbon-neutral energy sources opens new possibilities for the use of $CO_2$ as a medium for energy storage and as a source for the production of building blocks in a future fossil fuel-free chemical industry[1,2]. However, this vision faces the key challenge of developing highly active and stable electrocatalysts capable of targeting a single reduction product and of inhibiting the competing hydrogen evolution reaction (HER) in aqueous media[3]. The emergence of synergistic effects in multicomponent catalysts, such as alloys[4–9] and supported metal nanoparticles[10,11], provides unique opportunities to overcome the limitations of transition metals typically used for this reaction[12], whose performance is inherently constrained by the scaling relation between the binding energies of intermediates[13,14]. However, there is a need to derive clear structure–performance relationships that can guide the optimization of these complex materials and achieve breakthrough advances in an efficient manner.

Indium-modified copper-based materials are a promising family of multicomponent catalysts for $CO_2$ reduction, as evidenced by their enhanced performance for CO production over a wide variety of architectures[15–18]. Takanabe and colleagues[15,19] initially related the high selectivity toward CO of these catalysts to a HER-inhibiting effect from indium in alloys formed under reaction conditions (e.g., $Cu_{11}In_9$). In contrast, our previous studies on Ag-In[20] and Cu-In[21] catalysts revealed a synergistic interaction between metallic components and oxidic indium phases, suggesting the possibility of highly e$CO_2$RR-active bifunctional sites located at metal–oxide interfaces (instead of purely metallic compositions) being responsible for the enhanced performance of these indium-modified catalysts. In addition, the behaviour of these catalysts is marked by the occurrence of extensive structural and compositional changes upon exposure to reaction conditions[19,21]. In this context, the synthesis and characterization of materials with a high degree of structural and compositional control are key to deconvoluting this complex picture and obtaining insights into the nature of the active sites in these multicomponent catalysts.

In this direction, surface science studies have investigated the role of metal–oxide interfaces in the adsorption and activation of $CO_2$ over Au/CeO$_x$ catalysts[22], but the applicability of ultra-high vacuum techniques in the e$CO_2$RR is limited by their large gap with the actual environment of the electrochemical reaction. A way to bridge this gap is to use micro- and nanostructuring processes, which have been valuable for studying other photo- and electrochemical systems[23–25], to fabricate model electrodes that can be tested under e$CO_2$RR conditions and whose catalytic performance can thus be directly related to their structure and composition, as recently demonstrated (in single-component systems) by epitaxially grown Cu electrodes[26], and by Au and Cu catalysts with a controlled grain-boundary density[27–29].

Here, we demonstrate an experimental approach based on microfabrication that sheds light into the formation process and the nature of the active phase in copper-indium catalysts for $CO_2$ reduction. This approach consists in the deposition of microscale $In_2O_3$ or In structures, patterned by ultraviolet (UV) photolithography, on Cu or $Cu_2O$ surfaces. The relation between the catalytic activity toward CO of the structured electrodes and their geometry and composition, in combination with microscopy and elemental analyses facilitated by their regular microstructure, link the e$CO_2$RR activity of the electrodes to the formation upon exposure to reaction conditions of indium-poor bimetallic phases at the vicinity of the indium–copper interfaces. We remark that, compared with lithographic techniques with nanoscale resolution but limited throughput and accesibility[30–33], the use of UV photolithography is uncommon in model catalyst studies, likely

due to its limited size resolution (ca. 1 μm). However, this study shows how its high-throughput capability combined with careful consideration of the geometry in the design can overcome such limitations to derive catalytically valuable insights.

## Results

**Microfabrication and geometry of the structured electrodes.** Based on our previous work on indium-modified electrocatalysts, we initially hypothesized that the synergistic effect observed in these systems was due to the existence of bifunctional sites with high CO evolution activity located at the interfaces between oxidic indium phases and metallic components[20–22]. We further reasoned that, if this premise is true, a direct relationship between the CO evolution activity of these catalysts and the length of the interfaces between its components (e.g., $In_2O_3$ and Cu) would be observed. Its experimental confirmation thus required a synthetic procedure that provided a high degree of control over the catalyst composition and structure, and that yielded large electrodes suitable for testing under standard e$CO_2$RR conditions with an accurate measurement of reaction products[26]. Techniques from semiconductor device fabrication, such as physical vapour deposition processes and photolithography, comply with these requirements. Microfabrication thus allowed the production of several sets of electrodes with well-defined composition and geometry, which consisted of regular arrays of circular islands (comprising the indium-based component) deposited on a flat surface (the copper-based component). With this geometry, interfacial (and presumably highly active) sites would be located at the perimeters of the islands, whose total length can be defined by patterning via photolithography.

The main steps of the microfabrication process are summarized in Fig. 1a. In the first step, a Cu or $Cu_2O$ surface (ca. 200 nm) was deposited on a silicon wafer by direct current (DC) magnetron sputtering. A photoresist was then applied and patterned with UV light to expose microsized circular areas for the deposition of shallow $In_2O_3$ or In islands (ca. 90 nm) by electron beam evaporation, followed by the lift-off of the resist and dicing to obtain the structured electrodes (geometric total surface area, $A_{total} = 2.25$ cm$^2$). Three different compositions (denoted as island/surface) were prepared: (1) $In_2O_3$/Cu, (2) $In_2O_3$/$Cu_2O$, and (3) In/$Cu_2O$. The scanning electron microscopy (SEM) imaging of the fresh electrodes (Supplementary Fig. 1) showed that the Cu and $Cu_2O$ surfaces consisted of a dense agglomeration of nanometric crystallites (ca. 20 nm), resulting in broad reflections in the corresponding diffractograms. A similar structure was observed in the $In_2O_3$ and In islands, although their characteristic diffraction lines were not identified likely due to their amorphous nature (Supplementary Fig. 2).

The main geometrical design variable was the total length of the perimeters of the islands, which are periodically distributed on the electrode surface. This arrangement is characterized by a hexagonal two-dimensional unit cell in which the islands, with a diameter $d$, are separated by a pitch $c$ (Fig. 1b). We define the interfacial density $\rho_{int}$ as the total length of the perimeter of the islands per unit area of the electrode (see Methods). The number of islands per electrode ranged from about 25,000 for the lowest $\rho_{int}$ ($d = 50$ μm) to more than 40 million in the one with the highest ($d = 1.25$ μm). It is important to remark that the actual length of the interfaces should be proportional to $\rho_{int}$ with an influence from other factors (e.g., surface roughness). As remarked previously, we hypothesized that sites at the interfaces would be the main source of CO evolution activity. However, non-interfacial sites located at the exposed surfaces of each component would expectedly contribute to the e$CO_2$RR as well. To account for this, we aimed to achieve the same background

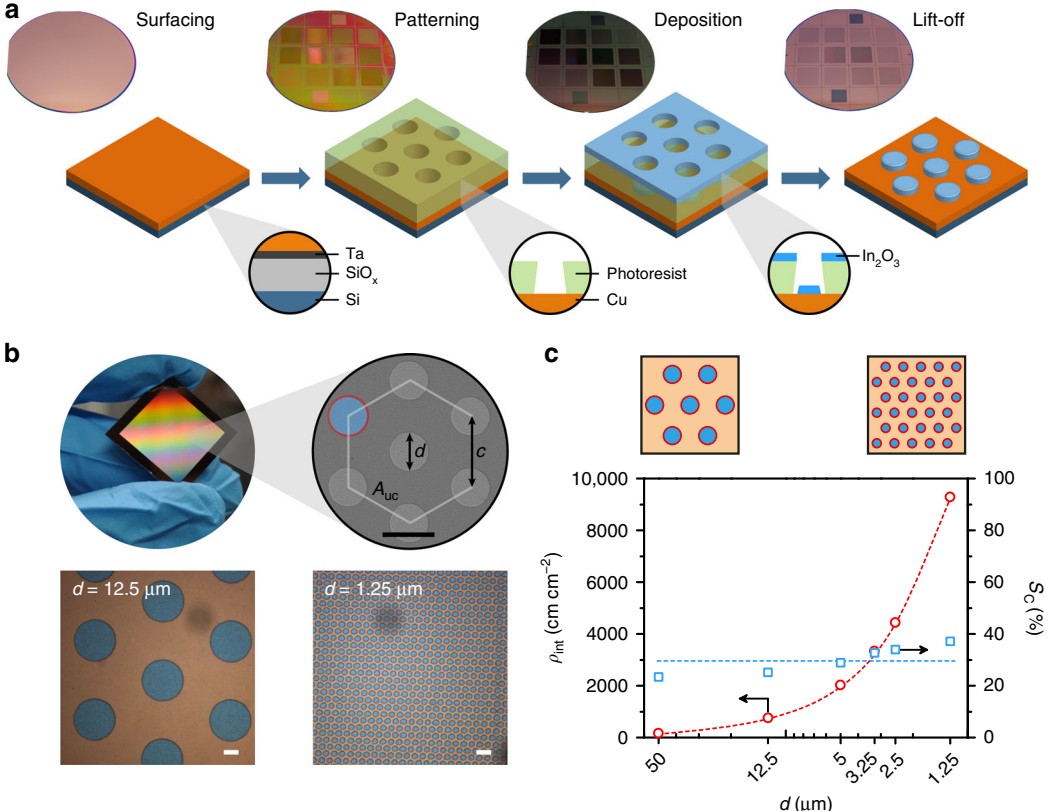

**Fig. 1** Microfabrication and geometry of the structured electrodes. **a** Scheme of the microfabrication of the structured electrodes (In$_2$O$_3$/Cu shown) with an image of the wafer at each main step: creation of a Cu surface on a Si wafer by sputtering (the underlying SiO$_x$ and Ta layers act as electrical and diffusion barriers, respectively), patterning of the sacrificial photoresist layer, deposition of In$_2$O$_3$ by electron beam evaporation, and lift-off of the photoresist. **b** Image of an individual structured electrode (In$_2$O$_3$/Cu, size 2 cm × 2 cm including edges, geometric active area $A_{total}$ = 2.25 cm$^2$) and a SEM micrograph showing the distribution of the islands on the surface with a hexagonal unit cell with area $A_{uc}$. The geometry of the arrangement is defined by the diameter of the islands ($d$) and the pitch ($c$). One of the islands and its interfacial perimeter are highlighted in blue and red, respectively. Optical micrographs show the size differences of the islands of two In$_2$O$_3$/Cu electrodes. Scale bars: 5 μm. **c** Actual interfacial density ($\rho_{int}$) and fraction of the electrode surface ($S_C$) covered by In$_2$O$_3$ (or In), measured by optical microscopy after the microfabrication of the model electrodes, as a function of the nominal island diameter ($d$). Trendlines are added as a visual aid

activity (i.e., not originating from the interfaces) over all electrodes by splitting the total geometric surface area between the components always in the same way. This is achieved by adopting the same $d/c$ ratio in all geometries. For this study, we defined $d/c = 0.5$, which would result in a coverage of the surface ($S_C$) with In$_2$O$_3$ or In of ca. 25% irrespective of $d$ and $\rho_{int}$ (see Methods). Figure 1c shows the actual interfacial density and the covered surface of the structured electrodes, denoted by the nominal diameter of the islands. Actual diameters were slightly larger than the nominal ones, resulting in the slight variation of $S_C$ across different samples (range: 23–37%). However, this variation is small compared to the very large (intentional) difference in the $\rho_{int}$ (range: 180–9300 cm cm$^{-2}$), which spans two orders of magnitude.

**Catalytic activity**. The large geometrical active area of the structured electrodes allowed their direct electrochemical testing and the quantification of the reaction products with a standard electrochemical cell for eCO$_2$RR studies (Supplementary Fig. 3). A short reaction time is desirable to reduce the interference of bubbles that gradually accumulate on the electrode surface and to limit the extent of restructuring, so that any changes observed in the electrodes can be related to their initial state while still allowing the assessment of the catalytic activity[20,34]. Fully

representative results could be obtained after 5 minutes of electrolysis due to the rapid equilibration of both the current density and composition of the small cathodic headspace of the cell (ca. 2 cm$^3$), as evidenced by the chronoamperometric curves of longer electrolyses and the corresponding product analysis (Supplementary Fig. 4).

We started by considering the In$_2$O$_3$/Cu system. The catalytic testing revealed that the HER was highly favoured over the eCO$_2$RR, as reflected by the partial current densities toward CO ($j_{CO}$) and H$_2$ ($j_{HER}$) shown in Fig. 2. In fact, all the In$_2$O$_3$/Cu electrodes showed a similar $j_{CO}$ (ca. −25 μA cm$^{-2}$) independently of the interfacial density. Interestingly, the control Cu electrode (i.e., lacking any In$_2$O$_3$ islands) showed a similar $j_{CO}$ and $j_{HER}$ to the In$_2$O$_3$/Cu electrodes, while a continuous In$_2$O$_3$ film showed comparatively low activity for both CO and H$_2$ evolution, in line with reported findings[20,35]. From these observations, no synergistic effect favourable for CO evolution in the In$_2$O$_3$/Cu system was apparent. In this regard, the small CO evolution activity evidenced in the In$_2$O$_3$/Cu electrodes most likely originated from the uncovered Cu surface. Notwithstanding the lack of influence of $\rho_{int}$ on the $j_{CO}$ in the In$_2$O$_3$/Cu system, the $j_{HER}$ did show a decrease with $\rho_{int}$ which is evident for island sizes smaller than 5 μm. However, this result is not accompanied by significant changes in the current efficiency for H$_2$ (ca. 70%, Supplementary Fig. 5). The reasons for this behaviour are not immediately clear,

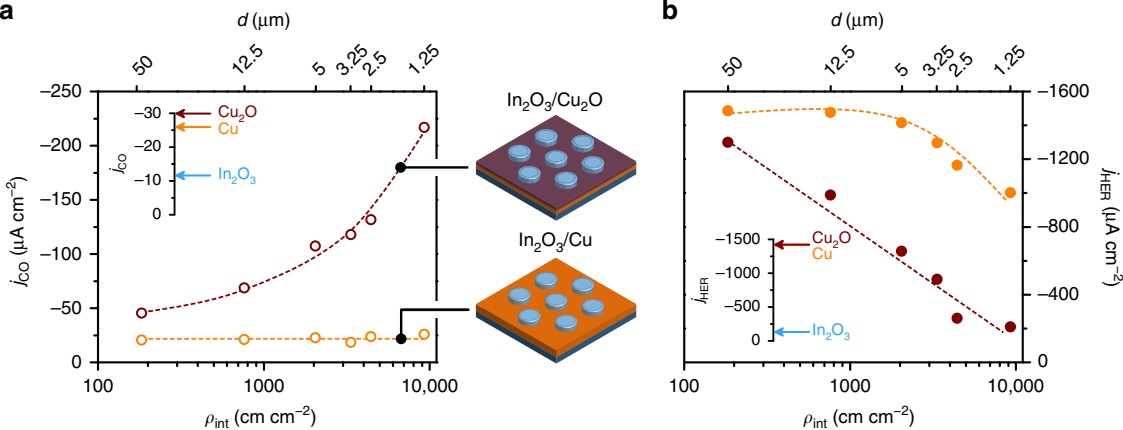

**Fig. 2** Catalytic activity for CO and $H_2$ evolution over structured $In_2O_3/Cu_2O$ and $In_2O_3/Cu$ electrodes. Partial current density for **a** CO ($j_{CO}$) and for **b** $H_2$ ($j_{HER}$) in $eCO_2RR$ electrolyses as a function of the interfacial density ($\rho_{int}$) over $In_2O_3/Cu$ and $In_2O_3/Cu_2O$ structured electrodes. The corresponding island diameter ($d$) is also indicated and the trendlines are added as a visual aid. The reaction was carried out in $CO_2$-saturated 0.1 M $KHCO_3$ (pH 6.7) at −0.6 V vs. RHE. The insets indicate the corresponding partial current densities measured over the single-phase control samples

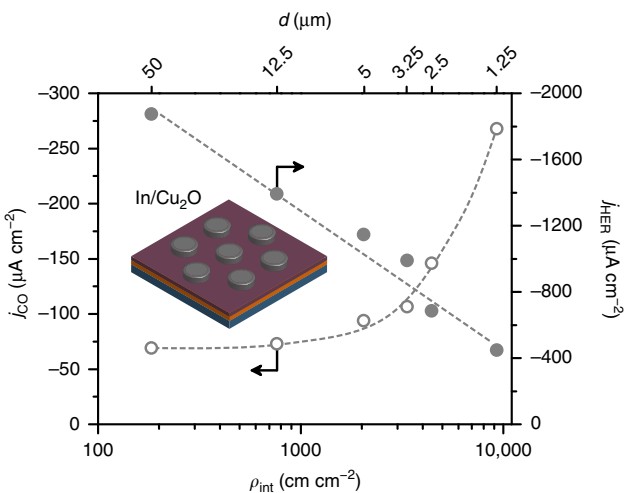

**Fig. 3** Catalytic activity for CO and $H_2$ evolution over structured $In/Cu_2O$ electrodes. The partial current density for CO ($j_{CO}$) and for $H_2$ ($j_{HER}$) over $In/Cu_2O$ electrodes follow a similar trend with the interfacial density ($\rho_{int}$) as over $In_2O_3/Cu_2O$ electrodes, indicating that In and $In_2O_3$ act as equivalent sources of indium for the formation of the active phase on $Cu_2O$ substrates upon exposure to reaction conditions

but the decrease of $j_{HER}$ might be related in part to the increased covering of the highly HER-active Cu surface with much less active $In_2O_3$ in the electrodes with the smaller island sizes (Fig. 1c).

The combination of the two oxides in the $In_2O_3/Cu_2O$ electrodes resulted in a very different picture. Over the electrodes with the lowest interfacial density (i.e., with $d = 50$ μm), $j_{CO}$ and $j_{HER}$ were comparable to the controls and to $In_2O_3/Cu$. However, in stark contrast to the latter, the increase of the interfacial density in the $In_2O_3/Cu_2O$ electrodes resulted in a marked enhancement of the catalytic activity toward CO (Fig. 2a). The relationship between $j_{CO}$ and $\rho_{int}$ clearly evidences the existence of a synergistic effect in $In_2O_3/Cu_2O$ electrodes that is linked to the interfaces. Since this behaviour was not observed when $In_2O_3$ islands were deposited on metallic Cu substrates, this result indicates that the presence of $Cu_2O$ (which is reduced under $eCO_2RR$ conditions to form oxide-derived Cu) is required for this effect to occur. A $Cu_2O$ electrode without any islands also showed

higher $eCO_2RR$ activity than the corresponding metallic Cu electrode, in line with recent reports on the activity-enhancing features of oxide-derived Cu (OD Cu), such as its higher density of grain boundaries[29,36]. However, the marked increase of $j_{CO}$ observed upon the deposition of $In_2O_3$ islands on $Cu_2O$ substrates cannot be explained by such effects alone. We remark that small amounts of formate were produced during the electrolyses over $In_2O_3/Cu_2O$ electrodes, with a current efficiency of 4–7% (Supplementary Fig. 6). However, this figure did not follow any clear trend with the interfacial density. In addition, the large fraction of the total charge transferred during the electrolyses that do not correspond to the measured reaction products is an indication of the reduction of $In_2O_3$ and $Cu_2O$ under reaction conditions.

Following this, we investigated the effect of substituting $In_2O_3$ with metallic In on the $Cu_2O$ surface, as this would provide an indication of the role of oxidic indium species in the enhancement of the CO evolution activity. The $In/Cu_2O$ system showed very similar results to $In_2O_3/Cu_2O$, in terms of both the activity toward CO and $H_2$ evolution and the dependence of $j_{CO}$ on $\rho_{int}$ (Fig. 3). The similarity between these two systems indicates that $In_2O_3$ acts as a source of metallic In upon reduction under $eCO_2RR$ conditions, in line with previous studies[20,21,37]. More tellingly, the fact that the same behaviour and performance was observed when the oxygen-rich indium source was substituted with an oxygen-poor one strongly suggests that sites with enhanced $eCO_2RR$ activity contain metallic (rather than oxidic) indium species under reaction conditions.

It should be noted that the increase of $j_{CO}$ with $\rho_{int}$ in $In_2O_3/Cu_2O$ and $In/Cu_2O$ electrodes was accompanied with a strong suppression of the HER (Figs. 2 and 3), despite the fact that the uncovered $Cu_2O$ surface (which is reduced to Cu under $eCO_2RR$ conditions and is highly HER active, as shown in the inset of Fig. 2b) occupies a similar fraction of the surface area of all electrodes (Fig. 1c). Taking this into account, if the $eCO_2RR$-active sites were located exactly at the perimeters of the islands (as originally hypothesized) this would indeed result in the increase of $j_{CO}$ with $\rho_{int}$ but not necessarily in an inverse relationship between $j_{HER}$ and $\rho_{int}$, assuming that the uncovered Cu surface retained its unmodified HER activity. Following this reasoning, the fact that both $j_{CO}$ and $j_{HER}$ are strongly related to $\rho_{int}$ indicates that the active sites formed by the interaction of the $Cu_2O$ substrate with $In_2O_3$ and In under reaction conditions are not confined to the initial interfaces. Instead, this result suggests that

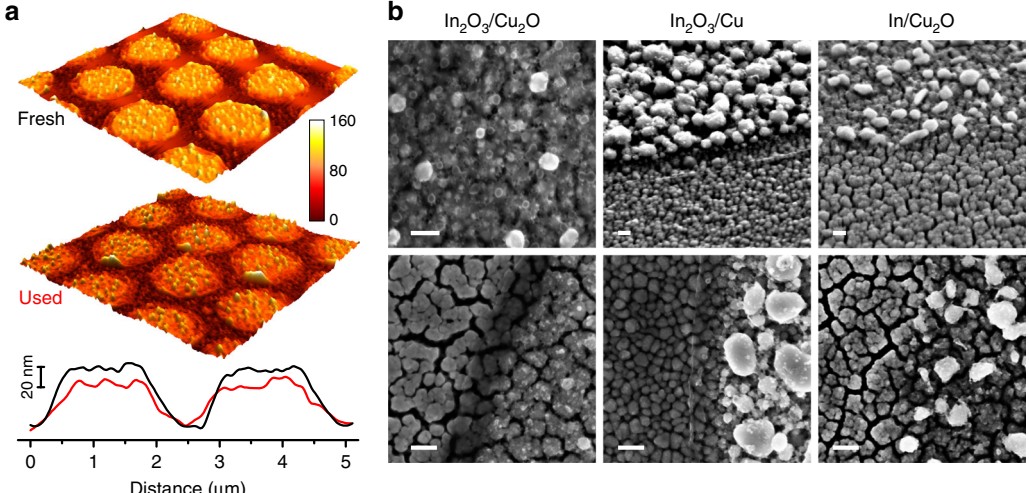

**Fig. 4** Microscopy of structured electrodes after the catalytic testing. **a** Pre- and post-reaction AFM analysis over a representative $In_2O_3/Cu_2O$ electrode ($d = 1.25\,\mu m$) and corresponding profiles across two adjacent islands. Analysed area ca. $7.2\,\mu m \times 7.2\,\mu m$. The colour scale is associated to nanometres. **b** Representative post-reaction SEM micrographs for different Cu-In compositions. The interior of the islands is distinguishable by the presence of white particles. The top left image shows the centre of and island. Scale bars: 100 nm

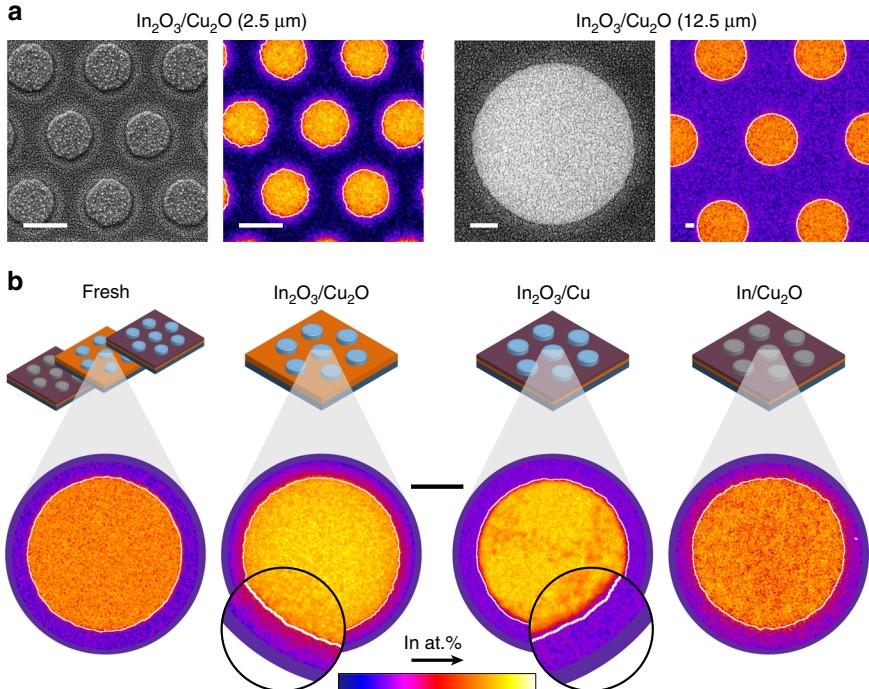

**Fig. 5** Elemental distribution of indium after the catalytic testing. **a** Representative micrographs in secondary (left) and backscattered (right) electron mode and distributions of indium obtained by SEM-EDX for electrodes with different island diameters (2.5 and 12.5 μm), showing the presence of an indium-rich concentric area after the reaction in $Cu_2O$-based electrodes. Scale bars: 2 μm. **b** Elemental distribution of indium over individual islands for different Cu-In compositions. A representative island prior to the reaction is shown as reference (Fresh). The white contours indicating the border of the corresponding fresh islands are added as a visual aid (see Supplementary Fig. 7). Scale bar: 5 μm

the interaction between the components modifies the $Cu_2O$-derived surface around the perimeters of the islands.

**Structural and compositional characterization.** Previous reports indicate that metallic copper and indium react to form intermetallic compounds (IMCs) with defined stoichiometry, even at room temperature[21,38]. In light of this fact, we examined the morphology and elemental distribution of selected structured

electrodes upon exposure to the reducing conditions of the $eCO_2RR$.

In terms of structure, atomic force microscopy (AFM) analysis evidenced the preservation of the overall geometry of the electrodes and a decrease of the height of the islands following the $eCO_2RR$ electrolysis (Fig. 4a). SEM imaging of the used electrodes revealed that material from the $In_2O_3$ and In islands still remained within the original perimeters after the reaction, and the Z-contrast of the micrographs in backscattered-electron

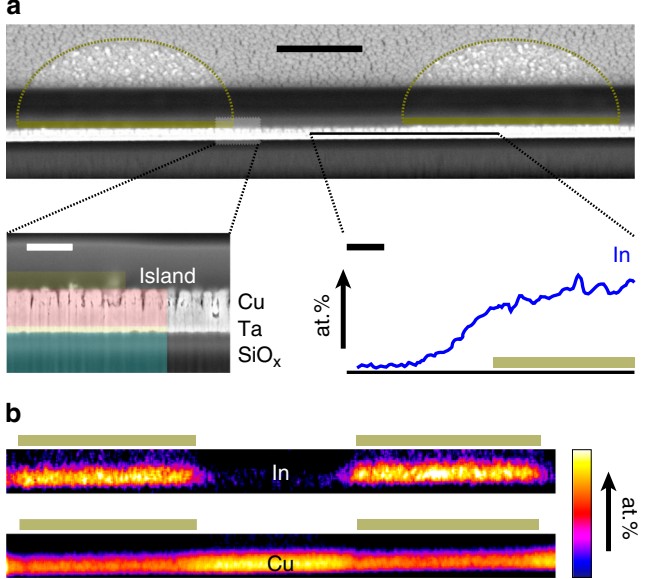

**Fig. 6** Cross-section analysis after the catalytic testing. **a** Cross-section of a representative $In_2O_3/Cu_2O$ ($d = 5\,\mu m$) electrode comprising two adjacent islands after the $CO_2$ reduction reaction obtained by FIB-SEM. Insets: multilayer structure (left) and variation of the concentration of indium along the indicated line obtained by EDX (right). Scale bars: 2 μm (main image), 400 nm (insets). **b** In and Cu elemental maps spanning the substrate and islands as shown in **a**. The borders of the islands and their corresponding position in the cross-sections are indicated in gold

detection (BSED) mode corroborated the distinct composition of this remaining material within the perimeter and of the substrate (Supplementary Fig. 1). However, the dense structure of the original islands was not conserved after the reaction. Instead, the islands in the used electrodes were composed of particles in a more sparse arrangement (Fig. 4b). The population of these In-rich particles within the perimeter of the islands appears to be higher in the used $In_2O_3/Cu$ electrode compared to the $Cu_2O$-based counterparts. The expected reduction of $Cu_2O$ to Cu was evidenced in the X-ray diffractograms of the used electrodes, but no indium-containing phases (e.g., $In_2O_3$, metallic In, or IMCs) could be identified likely as a consequence of a very low concentration (Supplementary Fig. 2).

The inspection of the electrodes by SEM at higher magnification revealed significant differences between the Cu and the $Cu_2O$ substrates after the reaction. Whereas the Cu surface appeared to retain its original (dense) morphology, the reduction of $Cu_2O$ to Cu resulted in a more irregular surface with a cracked appearance irrespective of the presence of the islands (Fig. 4b). The loss of oxygen from the $Cu_2O$ lattice upon reduction would explain the formation of voids, leading to an irregular structure of crystallites linked by grain boundaries[36,39].

The elemental distribution of Cu, In, and O in the fresh and post-reaction electrodes was analysed by energy-dispersive X-ray spectroscopy coupled to SEM (SEM-EDX, see Fig. 5). In the fresh electrodes, indium was expectedly found only within the perimeters of the islands. However, exposure of the $Cu_2O$-based electrodes to $eCO_2RR$ conditions provoked the appearance of a concentric area around the islands (i.e., a halo) visible both in secondary electron (SE) and BSED modes (Fig. 5a), reflecting a higher average atomic number compared to the rest of the Cu substrate. The corresponding elemental mapping (Fig. 5a) evidenced the presence of indium in these halos (and thus, outside the original perimeters of the islands). However, these

regions were absent in the poorly-performing $In_2O_3/Cu$ sample (Fig. 5b).

Although a reliable quantification was not possible, the EDX analysis indicated that the concentration of indium in the halos is very low compared to the islands (see scale bar in Fig. 5). This observation points to the diffusion of indium from the islands upon exposure to reaction conditions as the most plausible mechanism for their formation. To shed light on this phenomenon, we analysed the elemental distribution of In and Cu in cross-sections of a post-reaction $In_2O_3/Cu_2O$ electrode by focused ion beam coupled with SEM (FIB-SEM). Figure 6a shows the cross-section of an electrode comprising two adjacent islands. The EDX analysis of the copper substrate and the islands (Fig. 6b) unequivocally showed the diffusion of indium across the entire depth of the substrate under the islands upon reduction of $In_2O_3$. Interestingly, an incipient lateral diffusion toward the exterior of the islands was also apparent, thus confirming the three-dimensional character of the observed halos and strongly pointing to a Cu-In solid-state reaction (as opposed to a dissolution-deposition process from the electrolyte, for example) as the mechanism behind their formation. A linescan analysis across an island complemented these results (Fig. 6a), revealing the variation of the concentration of indium within the halos and confirming a typical width of ca. 1 μm (Fig. 5).

Time-of-flight secondary ion mass spectrometry (ToF-SIMS) and X-ray photoelectron spectroscopy (XPS) provided additional insights into the nature of In species present on the surface of $In_2O_3/Cu_2O$ electrodes after the reaction. The distribution of CuIn and InOH ions obtained by ToF-SIMS confirmed the formation of IMCs and indium hydroxide, respectively, within the islands (Supplementary Fig. 8). By XPS we aimed to compare the chemical state of surface species at the centre, the perimeter, and the exterior of the islands ($d = 50\,\mu m$) by careful positioning of the X-ray beam. The analysis of the Cu and In regions did not allow the identification of any distinct spectroscopic fingerprints associated to the halo (Supplementary Fig. 9), although this is complicated by the large diameter of the X-ray beam of conventional XPS equipment compared to the very narrow geometry of the halo region (see Methods for more details). Nevertheless, analysis of the O 1 s signal confirmed the formation of hydroxide on the surface, especially within the islands. In view of the reductive conditions dominant under the $eCO_2RR$, this hydroxide is expected to originate after the return of metallic indium to open circuit potential after the electrolysis and/or upon exposure to air, as observed in a previous work[21].

The presence of the halos in the $Cu_2O$-based electrodes (and their absence in $In_2O_3/Cu$) initially suggested that the diffusion process is much more favourable in the irregular Cu layer formed upon the reduction of $Cu_2O$ than in the denser and more regular Cu film in $In_2O_3/Cu$. This could be a consequence of distinct features of the oxide-derived Cu layer, such as favoured diffusion of In through grain boundaries[38]. To investigate this, we studied electrodes composed of metallic In islands deposited directly on an oxide-derived Cu substrate (In/OD Cu). This substrate was prepared by the electrochemical reduction of a $Cu_2O$ layer prior to the photolithographic process (see Methods for details). Once prepared, the electrodes were stored under ambient conditions for 6 days and then analysed by SEM-EDX (Fig. 7a) to evaluate the possibility of In diffusion through OD Cu under non-electrochemical conditions. Micrographs showed the familiar irregular appearance characteristic of the OD Cu surface (see Fig. 4b). However, there was no evidence of indium diffusion toward the exterior of the islands. Interestingly, the same result was observed when In/OD Cu electrodes (stored under the same conditions) were subjected to $eCO_2RR$ electrolysis (Fig. 7a). Taken together, these results strongly suggest that the rapid

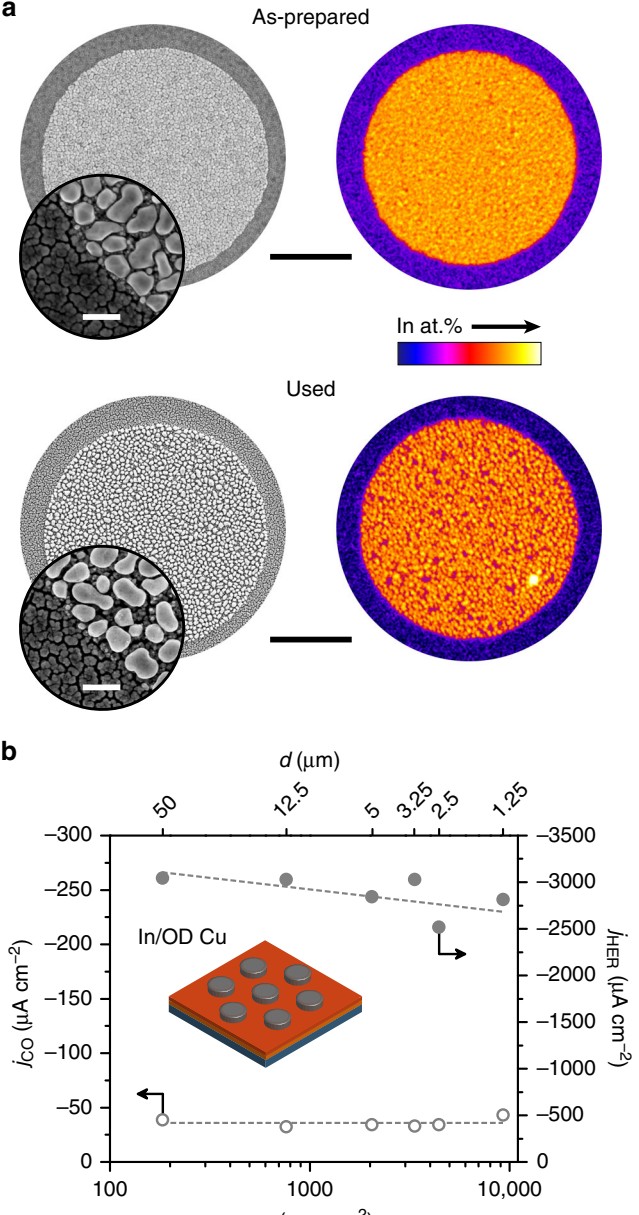

**Fig. 7** Microscopy and elemental distribution of indium, and catalytic activity for CO and $H_2$ evolution over structured In/OD Cu electrodes. **a** Representative SEM micrographs and EDX chemical map of indium of (top) as-prepared electrodes ($d = 12.5\,\mu m$) after 6 days stored under ambient conditions and (bottom) upon reaction. Scale bars: 5 μm (main images), 200 nm (insets). **b** Partial current density for CO ($j_{CO}$) and for $H_2$ ($j_{HER}$) in eCO$_2$RR electrolyses as a function of the interfacial density ($\rho_{int}$). The corresponding island diameter ($d$) is also indicated and the trendlines are added as a visual aid. The reaction was carried out in CO$_2$-saturated 0.1 M KHCO$_3$ (pH 6.7) at −0.6 V vs. RHE

diffusion of indium to form the halos observed in Cu$_2$O-based electrodes is actually caused by the reduction of the Cu$_2$O phases in the presence of indium species upon exposure to eCO$_2$RR conditions. In addition, the CO evolution activity of In/OD Cu electrodes was generally poor (Fig. 7b) and unrelated to the interfacial density, resembling results from In$_2$O$_3$/Cu electrodes (Fig. 2) that did not form a halo region, either. Nevertheless, we remark that additional studies on the solid-state diffusion kinetics

of the Cu-In system under eCO$_2$RR conditions would be necessary to further rationalize this behaviour.

In this context, the indium-modified copper surfaces that form the halos emerge as the most likely origin of the eCO$_2$RR activity (and HER suppression) of the structured Cu$_2$O-based electrodes. Because these concentric areas expand outwards from the islands perimeters, they would modify the highly HER-active Cu surface with sites that are presumably very active toward CO production (and much less toward H$_2$ evolution). The fraction of the electrode surface modified by this effect depends on the diameter of the islands and on the width of the concentric rings, but essentially it follows the variation of $\rho_{int}$ (see Methods). Further support for this mechanism was obtained by studying geometrically similar SnO$_2$/Cu$_2$O electrodes, which also showed enhanced activity toward CO. In this case, EDX maps displayed a longer range modification of the copper substrate by tin, leading to the complete modification of the uncovered surface among islands (Supplementary Fig. 10), and concomitantly, to an eCO$_2$RR activity which was independent of the $\rho_{int}$.

The results herein presented suggest that the modification of copper with a low amount of indium induces high activity toward CO evolution. This echoes findings from In-modified Cu$_2$O powder catalysts (i.e., ca. 5 at.% In)[17], as well as recent work by Berlinguette and colleagues[16] who found an optimal composition for CO activity (Cu$_3$In) that is richer in Cu than previously reported by Takanabe and colleagues[15]. Taken together, these observations indicate that metallic In-modified Cu phases, rather than metal–oxide interfaces as we had initially hypothesized[21], are the main source of eCO$_2$RR activity toward CO in Cu-In catalysts. Nevertheless, it is possible that In(OH)$_3$, which is an electrochemically silent species under studied conditions[37], plays a beneficial role in catalyst performance by acting as a local indium capture and storage system. As revealed by ToF-SIMS and XPS (Supplementary Fig. 8 and 9), it is likely that part of the metallic In that penetrates into Cu is transformed into the inert hydroxide form upon exposure to open circuit potential and/or moisture. Because this hydroxide cannot be reduced back to metallic In at eCO$_2$RR potentials[20,21,37], a succession of reduction/oxidation cycles, such as those applied in our previous work[21], would aid the progressive formation of (highly eCO$_2$RR-active) indium-poor metallic compositions by accumulating indium into a In(OH)$_3$ sink. However, we remark that the existence of active sites at the interfaces between In(OH)$_3$ and metallic phases cannot be discarded at this point.

## Discussion

From a broad perspective, this work demonstrates that the controlled variation of the geometrical and compositional characteristics of microfabricated electrodes can be used to derive its correlations with catalytic performance, monitor dynamic processes, and gain insights into the nature of the active phase in multicomponent electrocatalysts. Moreover, the microscale nature of the electrodes is favourable for their integration with in situ and operando characterization techniques. In particular, the Cu-In system was studied by the preparation, evaluation under eCO$_2$RR conditions, and characterization of sets of electrodes consisting of In$_2$O$_3$ or In circular islands regularly distributed on flat Cu or Cu$_2$O surfaces. The design of these structured electrodes achieved a variation of the density of the interfaces between the components across several orders of magnitude. A direct relationship between this parameter and the CO evolution activity (with concurrent suppression of the parasitic HER) was disclosed when the reduction of the Cu$_2$O substrate and the eCO$_2$RR occurred simultaneously, irrespective of the initial phase of indium. Electron microscopy, EDX, and ToF-SIMS analyses

related this observation to the formation via a solid-state diffusion mechanism of highly active indium-modified regions around the initial interfaces upon exposure to reaction conditions. This study sheds light into the mechanism of formation of the active phase that results in the enhanced performance toward CO commonly observed over copper-indium catalysts with respect to the pristine Cu counterpart, most likely composed of bimetallic compositions with very low amounts of indium as per the abovementioned results. We anticipate that the use of microfabrication to produce structured electrodes with geometric and compositional control that can be tested under relevant operating conditions, combined with the use of spatially resolved characterization techniques, will contribute to bridging the gap between model systems and surface science studies for the investigation of synergistic effects in multicomponent electrocatalysts.

## Methods

**Geometrical definitions**. The islands are distributed on the geometrical active surface of each electrode ($A_{total} = 2.25$ cm$^2$) in a periodic hexagonal array. This arrangement is defined by a two-dimensional unit cell in which islands of diameter $d$ are separated by a pitch $c$ (i.e., the centre-to-centre separation), as shown in Fig. 1b. Consequently, the area of this unit cell $A_{uc}$ is given by Eq. (1).

$$A_{uc} = \frac{3\sqrt{3}}{2}c^2. \tag{1}$$

The total perimeter of the islands within the unit cell is three times their circumference:

$$L_{uc} = 3\pi d. \tag{2}$$

The interfacial density $\rho_{int}$ is defined as the total length of the perimeter of the islands per unit area of the electrode. Therefore, $\rho_{int}$ can be calculated as the ratio of $L_{uc}$ to $A_{uc}$, as shown in Eq. (3). It follows that this value can also be obtained by multiplying the total number of islands on the electrode $n$ by their circumference, and then dividing by the electrode area.

$$\rho_{int} = \frac{L_{uc}}{A_{uc}} = \frac{2\pi\sqrt{3}}{3}\left(\frac{d}{c^2}\right) = \frac{n\pi d}{A_{total}}. \tag{3}$$

Analogously, the fraction of the electrode surface covered by the In$_2$O$_3$ or In islands ($S_C$) can be calculated either from the unit cell or from the total surface of the electrode, as shown in Eq. (4).

$$S_C = \frac{A_{island,uc}}{A_{uc}} = \frac{\pi\sqrt{3}}{6}\left(\frac{d}{c}\right)^2 = \frac{n\pi d^2}{4A_{total}}. \tag{4}$$

The fraction of the electrode surface modified by the diffusion of indium from the islands ($S_M$) depends on the diameter of the islands and on the width of the concentric rings ($x$), but more crucially, on the total number of islands on each electrode, which varies in the same way across different samples as the interfacial density, as shown in Eq. (2). In geometric terms, the relationship between these variables is shown by Eq. (5).

$$S_M = \frac{n\pi}{A_{total}}(x^2 + xd) = \rho_{int}\left(\frac{x^2}{d} + x\right). \tag{5}$$

**Substrate preparation**. 100 mm Si(100) wafers were used as substrates for the microfabrication process. In the first step, a SiO$_x$ layer (500 nm thickness) was deposited on each wafer by plasma-enhanced chemical vapour deposition in an Oxford Instruments Plasmalab 80 Plus system (conditions: 300 °C, 900 mTorr). The SiO$_x$ layer served to electrically isolate the electrode surface from the semiconducting substrate. Afterward, successive layers of tantalum (50 nm) and copper (250 nm) were deposited on the SiO$_x$/Si wafers by DC magnetron sputtering of the respective metal targets (MaTeCK GmbH, 99.95% Ta and 99.99% Cu, diameter 76.2 cm) in a Mantis HiPIMS deposition system equipped with confocal sources (conditions: target-to-substrate distance of ca. 350 mm, chamber pressure of 8 mTorr, no substrate heating, 80 cm$^3$ STP (standard temperature and pressure) min$^{-1}$ Ar, currents of 400 and 300 mA for Ta and Cu, respectively, corresponding to ca. 115 and 130 W). The Ta layer acts as a diffusion barrier between Cu and SiO$_x$/Si. Under the aforementioned conditions, both layers were deposited at a rate of ca. 6 nm min$^{-1}$. For the Cu$_2$O-based electrodes, a Cu$_2$O layer (150 nm) was deposited on the metallic Cu film (150 nm) by reactive DC magnetron sputtering (conditions: chamber pressure of 8 mTorr, no substrate heating, 80 cm$^3$ STP min$^{-1}$ Ar, 9 cm$^3$ STP min$^{-1}$ O$_2$, 250 mA) at a rate of ca. 6 nm min$^{-1}$. The thickness and deposition rate of the films were monitored with a quartz crystal microbalance.

**Photolithography**. Before the application of the photoresist layers, the Cu substrates were cleaned in a dilute citric acid solution (1 wt.%) for 30 s to remove surface oxides, copiously rinsed with ultrapure deionized (DI) water, blow-dried with a N$_2$ gun, and finally dehydrated at 110 °C for 5 min on a hotplate. Cu$_2$O substrates were not cleaned in citric acid but otherwise processed in the same manner. A thin layer (ca. 300 nm at 3500 rpm) of LOR 3 A lift-off resist (Micro-Chem) was then spin-coated on the substrate and baked at 150 °C for 10 min on a hotplate, followed by the application of a thicker layer (ca. 2 μm at 3000 rpm) of AZ nLOF 2020 negative photoresist (Merck Performance Materials). This bi-layer process greatly simplified the removal of the photoresist after the deposition of the islands and avoided any reactions between AZ nLOF 2020 and the underlying Cu-containing surface. The photoresist was soft-baked at 110 °C for 60 s and exposed to UV through a chrome/soda-lime glass photomask (exposure dose: 55 mJ cm$^{-2}$ at the 365 nm wavelength) in a Karl Suss MA6 mask aligner. The photomask was designed as to define 12 patterned regions on the wafer, each including a dense hexagonal array of circles, which allowed the simultaneous fabrication of several electrodes with varying geometry in addition to control samples with either total or null covering. Following the exposure, the photoresist underwent a post-exposure bake at 110 °C for 60 s and was developed for 50 s in 2.38% tetramethyl ammonium hydroxide (AZ 826 MIF developer, Merck Performance Materials), thus exposing the microsized circular areas for the deposition of In$_2$O$_3$ or In islands.

**Oxide deposition and lift-off**. After the development of the photoresist, the deposition of In$_2$O$_3$ or In on the substrates (90 nm) at a rate of ca. 6 nm min$^{-1}$ was carried out in a Leybold Univex 500 electron beam evaporation system fitted with a 10 keV electron gun. The evaporation source was a graphite crucible filled with sintered In$_2$O$_3$ granules or In pellets (Kurt J. Lesker, 99.99%). The applied currents were low (<10 mA) to reduce heating of the evaporation chamber. The removal of the sacrificial photoresist layer (resulting in the lift-off of the excess In$_2$O$_3$) was carried out by dipping the wafer in $N$-methyl-2-pyrrolidone (NMP) at 80 °C for 60 min. The wafer was then transferred to a beaker with clean NMP, washed copiously with DI water, and blow-dried with a N$_2$ gun. The individual electrodes (2 cm × 2 cm) were obtained by dicing the wafer in a Disco DAD 321 dicing saw. A protective layer of mr-PL 40 photoresist (Micro Resist Technology GmbH, ca. 7 μm at 3000 rpm, soft-baked at 110 °C for 60 s) was applied to the wafer prior to dicing, which was then fully removed from the electrodes by cleaning them in NMP at 50 °C for a few minutes followed by copious rinsing with DI and blow-drying with a N$_2$ gun.

**Preparation of In/OD Cu electrodes**. To assess the influence of OD Cu on the diffusion of indium, an additional set of electrodes was prepared by depositing indium islands on a substrate whose initial Cu$_2$O layer had been previously reduced. To this end, a 4-inch wafer was first processed as previously described to produce a Cu$_2$O substrate. A large puncture-resistant polypropylene container (Braun Medibox) with a total volume of 3 dm$^3$ served as a one-compartment cell for the electrochemical reduction of the Cu$_2$O layer. The wafer was placed vertically in the cell with a Teflon wafer holder and the cell was filled with sufficient electrolyte (CO$_2$-saturated 0.1 M KHCO$_3$) to fully submerge the substrate. CO$_2$ was continuously bubbled into the electrolyte at a flow rate of 20 cm$^3$ STP min$^{-1}$. A large carbon gas diffusion layer (Sigracet 39BC, SGL Carbon) placed in close proximity (ca. 1 cm) and parallel to the substrate served as the counter electrode. Electrical contact with the working and counter electrodes was achieved with conductive copper foil tape (3 M). The leak-free Ag/AgCl reference electrode (3.4 M KCl, model LF-1, Innovative Instruments) was placed in the interelectrode space. The Cu$_2$O layer was reduced under similar conditions to the CO$_2$ reduction tests by holding the potential at −0.6 V vs. reversible hydrogen electrode (RHE) for 5 min. Cu$_2$O was fully reduced within 2 min to form a homogeneous oxide-derived copper layer, as confirmed by visual inspection and SEM analysis. The OD Cu substrate was then rinsed copiously with DI water and subsequent processing (e.g., photolithography, deposition of indium islands) was performed in the same manner as with a regular Cu wafer, as described.

**Electrode characterization**. AFM of the electrodes was carried out in a Bruker FastScan Dimension AFM. XRD patterns of the electrodes were obtained in a PANalytical X'Pert PRO-MPD diffractometer with Bragg-Brentano geometry using Ni-filtered Cu Kα radiation ($\lambda = 0.1541$ nm). The instrument was operated at 40 mA and 40 kV and the patterns were recorded in the 30–65° $2\theta$ range with an angular step size of 0.05° and a counting time of 5 s per step. SEM and EDX spectroscopy maps from the surface and internal cross-sections of the electrodes were acquired on a FEI Quanta 200F instrument equipped with an Ametek EDAX Octane Super detector. The relatively high density of copper and of indium in combination with a low accelerating voltage (5 kV) allowed to minimize the penetration depth of the incident beam (estimated as 100–200 nm) and thus provided elemental maps which are representative of the surface state. The low concentration of indium required long acquisition times (ca. 3 h) to obtain high signal-to-noise ratios. XPS analysis of an In$_2$O$_3$/Cu$_2$O electrode ($d = 50$ μm) following the reaction was carried out in a Physical Electronics Quantum 2000 spectrometer equipped with a 180° spherical capacitor energy analyser, at a base pressure of $5 \times 10^{-7}$ Pa using monochromatic Al Kα radiation (1486.68 eV) with a focused beam for small-area measurements (nominal spot size of 20 μm).

Spectra were acquired at a beam power of 4.5 W after centring the X-ray beam on three different locations of the electrode, corresponding to the bare substrate, the interface, and an island. Charge neutralization during the analysis was achieved by the simultaneous operation of electron- and argon ion-neutralizers at 1.2 and 10 eV, respectively. The binding energy scale was calibrated by setting the aliphatic component of the C 1 s peak to 284.8 eV during data processing. ToF-SIMS analysis of an $In_2O_3/Cu_2O$ electrode ($d = 12.5$ μm) following the reaction was carried out in the Physical Electronics TRIFT II instrument equipped with a Ga liquid metal ion gun. During analysis a $Ga_1^+$ primary ion beam in combination with an electron flood gun for charge compensation was scanned over the surface. Positive secondary ions were extracted and analysed with nominal resolution according to their respective charge-to-mass ratios. Cu and In maps comprise the combined signals from $^{63}Cu^+ + ^{65}Cu^+$ and $^{113}In^+ + ^{115}In^+$, respectively. Image treatment was performed with ImageJ software (Wayne Rasband, National Institutes of Health, USA, version 1.51). Contrast enhancement (0.1% saturated pixels) with equalized histogram was applied to SEM micrographs. EDX elemental maps underwent the same treatment and are represented in a Look-up Table (LUT) comprising from (R:28; G:0; B:134) to (R:255; G:255; B:255).

**Electrochemical tests**. A custom gastight sandwich-type cell with two compartments separated by a Selemion AMV anion exchange membrane (AGC Engineering) was employed for all electrochemical experiments, using a carbon gas diffusion layer (Sigracet 39BC, SGL Carbon) as the counter electrode and a leak-free Ag/AgCl reference electrode (3.4 M KCl, model LF-1, Innovative Instruments). A 0.1 M $KHCO_3$ solution (Sigma-Aldrich, 99.95% trace metals basis) prepared with 18.2 MΩ cm ultrapure water was used as the electrolyte. The cathodic and anodic compartments contained 9.5 and 7.8 cm³ of the electrolyte, respectively, which were saturated with $CO_2$ (Messer, purity 4.8) for at least 20 min prior to the start of the electrolysis, with a resulting pH of 6.75. $CO_2$ was continuously bubbled separately into both chambers during the electrolysis at a flowrate of 20 cm³ STP min⁻¹. All electrochemical measurements were carried out at room temperature with an Autolab PGSTAT302N potentiostat, and the potentials reported in this work are referred to the RHE scale. The potentiostatic electrolyses were carried out with the *iR* compensation function set at 80% of the uncompensated resistance $R_u$, which was determined before the start of the electrolysis by electrochemical impedance spectroscopy measurements at the reaction potential of −0.6 V vs. RHE. The recorded potentials were converted to the RHE scale following the electrolysis after manually correcting for the remaining uncorrected $R_u$. Following this correction, the applied potentials were within 10 mV of the target potential of the electrolysis.

**Product analysis**. For the analysis of the gas-phase reaction products, the outlet gas of the cathodic compartment of the cell flowed continuously through the sample loop of an SRI 8610C gas chromatograph (Multi-Gas #3 configuration) operating with Ar as carrier gas at a head pressure of 2.3 bar and equipped with HayeSep D and Molecular Sieve 13X packed columns. The partial current density $j_i$ for each gas-phase product (CO and $H_2$) during the electrolysis was calculated using Eq. (6), where $Q_{gas}$ is the molar flow of gas through the cell, $C_i$ is the molar concentration of the product in the outlet as determined by gas chromatography, $n$ represents the number of electrons transferred to form the product (i.e., 2 for CO and $H_2$), and $F$ is Faraday's constant. It follows that the current efficiency (CE$_i$) for each product is obtained by dividing $j_i$ by the recorded current at the sampling time $i_t$, as shown in Eq. (7). Gas flows in this work are referenced to normal conditions defined as 0 °C and 1.01325 bar.

$$j_i = Q_{gas}C_i nF, \tag{6}$$

$$CE_{i,gas} = \frac{j_i}{i_t} \times 100. \tag{7}$$

Very low concentrations of formate produced during the short $CO_2$ reduction electrolyses (i.e., lower than 100 μM) were quantified by nuclear magnetic resonance (NMR) spectroscopy in a Bruker Avance III HD 500 MHz spectrometer equipped with a CPPBBO probe. Following the reaction, catholyte and anolyte samples (700 μl) were mixed each with $D_2O$ (50 μl) containing phenol (50 mM, Acros Organics, 99%) as an internal standard. A calibration curve was obtained beforehand by analysing samples with known amounts of $HCOO^-$ in $CO_2$-saturated 0.1 M $KHCO_3$ and determining the relative integral of the $HCOO^-$ singlet ($\delta = 8.3$ ppm) normalized to that of the phenol triplet at $\delta = 7.2$ ppm. 1D $^1H$-NMR spectra with water suppression of the samples were recorded at 25 °C using the zgpr pre-saturation pulse program with the following parameters: 128 scans (NS) with a relaxation delay (D1) of 5 s at a power level (PL9) of 40.4 dB, pre-scan delay (DE) of 10 μs, acquisition time (AQ) of 4 s, and an 8000 Hz spectral width centered on the water peak (at ca. 4.7 ppm). These settings resulted in a high signal-to-noise ratio and an analysis time of ca. 20 min per sample. The calculation of the current efficiency for liquid-phase products (e.g., formate) is given by Eq. (8), where $V_e$ is the volume of the electrolyte, $C_i$ is the molar concentration of the product in the

electrolyte, and $q_{tot}$ is the total current passed during the electrolysis.

$$CE_{i,liquid} = \frac{V_e C_i nF}{q_{tot}} \times 100. \tag{8}$$

**Data availability**. The authors declare that the data supporting the findings of this study are available within the article and its Supplementary Information file. All other relevant source data are available from the corresponding author upon reasonable request.

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

## Acknowledgements

This work was sponsored by ETH Zurich (Research Grant ETH-01 14–1) and by the European Union through the A-LEAF project (732840-A-LEAF). The authors gratefully acknowledge the FIRST Center for Micro- and Nanoscience and the Scientific Center for Optical and Electron Microscopy (ScopeM) of ETH Zurich for access to their facilities. The authors thank J. Reuteler and M. Gabureac for their assistance with the FIB-SEM and AFM imaging of the structured electrodes, respectively; and J. Carroll (Photomask Portal) for his involvement and advice during the design of the photomask.

## Author contributions

J.P.-R. conceived and coordinated all the stages of this research. G.O.L. designed the electrode geometry, developed the microfabrication process, prepared the structured electrodes, and carried out catalytic tests. T.S. carried out part of the catalytic tests. A.J.M. carried out the characterization of the structured electrodes. All authors contributed to writing the manuscript.
