## [Peer Review File · Nature Communications]

Reviewers' comments:

Reviewer #1 (Remarks to the Author):

Authors carry out an investigation of CO₂ reduction at microfabricated electrodes, varying the composition and the In/Cu interface length to gain insights in the mechanism of formation of CO. The work is interesting and provides some insight into the mechanism, since the combination InO_x/CuO_y appears to perform better, suggesting an effect at the interface. Besides this insight however the mechanism of CO₂ reduction at Cu-In based systems does not appear to be better understood. The authors do not start with a mechanistic hypothesis to verify, but they only hypothesize an effect at the interface. This hypothesis in other works has been hypothesized and perhaps verified in terms of a bimetallic effect, which has been mentioned by the authors at the start of the manuscript. Overall, I do not think that this work provides sufficient insight for publication in this journal.

Additional comments follow:

- The abstract is long and at various places repetitive and not very clear; some statements are cryptic.
- The authors mention that they attempt to limit compositional changes of the catalyst by keeping the experiments short. In reality, Indium oxidation may occur at very short times, and In or In₂O₃ may transform in In(OH)_x.
- The surface diffusion during the reduction tests are due to gradients of electrochemical potential; these should be expected and their magnitude can be very different from conventional diffusion.
- the attempt to determine In surface diffusion should have been performed with a XPS, not with EDAX.

My interpretation of these data is that Cu and Indium interdiffuse to make a better catalyst due to the bimetallic effect. Cu and In tend to interdiffuse in the solid state as shown by several works showing successive electrodeposition of In over Cu, forming an alloy in a short time. As such, the work of Hoffman (ref. 22) shows a better way to maximize (but not control) the interface length between Cu and In.

Reviewer #2 (Remarks to the Author):

The manuscript provides insight into microfabrication techniques for fabrication of controlled electrodes in order to precisely evaluate the role of metal/metal-oxide catalysts on CO₂ reduction. The experiments are well designed and novel, and the results would be of interest for wide range of experts in electrocatalysis and microfabrication. The manuscript can be published with minor revision, after considering below comments:

1- "Abstract" and "conclusions" are a bit vague, please restructure and mention the core point and findings of your results.

2-The whole manuscript needs an English proof read. there are many long sentences, hard to follow and understand, specially in:
page 4 line 4; page 5 line 5-9; page 6 line 14-18, page 7 line 10-14; page 9 line 21-23; page 10 line 14-17; page 11 line 10-17.

3- Some Figures can be reorganized from supporting info to the main text to enhance reading and

understanding. For example supp. 1) Fig. 6 can be also added in Fig. 2.

Caption of Supp. Fig 1 is confusing. Supp. I suggest that Figure 1a can be moved to the main text to figure 1 to help understanding the microfabricated fresh electrode and their size difference. Figure 3b can be moved to the Supp. The description for Info. Supp. Fig. 5 is not clear well.

4- Supp. Figure 2: 1) correct the caption (-:6 V)

2) Please note the In₂O₃ peaks. 3) It would be also useful to add XRD of In/CuO₂ or In₂O₃/Cu (used) to investigate whether any phase transition occurred.

Reviewer #3 (Remarks to the Author):

This work by Javier Pérez-Ramírez et al., a precisely controlled interface density was created by the microfabrication technique, which provides an excellent platform to analyze the impacts of these interfaces. Suitable control experiments were relatively well conducted in combination with advanced characterization techniques. The proposed solid-state diffusion mechanism add new insight into the origin of activity on this bimetallic catalysts, and this strategy is promising in understanding the catalytic behaviors of other multicomponent catalysts. Overall, this manuscript was well written with persuasive results, and is potentially suitable for publication. However, a few minor issues need to be addressed before acceptance can be suggested.

1. "A short reaction time (5 min) was selected to reduce the interference of bubbles that gradually accumulate on the electrode surface and to limit the extent of restructuring, so that any changes observed in the electrodes could be related to their initial structure while still allowing the assessment of the catalytic activity." For most catalysts, the current would drop fast at the initial stage of electrolysis, and then reach semi-steady state after a few minutes, could the author show the current-time (i-t) curves? If the current decays quickly, the faradaic efficiency obtained at the beginning may not be accurate due to the head-space equilibrium issue. However, I realized a very small cell was used in this study which may have a small head space as well, possibly this issue can be alleviated.

In addition, since the total faradaic efficiency is only ~70%, can any liquid product be detected via H NMR? Formate is a possible liquid product in In-Cu systems. The estimated concentration is over 100 μM if assuming the remaining product is formate, which maybe below the detection limit of HPLC and IC, but is sufficient in NMR analysis. 128 scans with water suppression on a 500 MHz NMR are enough to get very good signal-to-noise ratio. The following parameters can be used: 0.5 s prescan delay, 5 s presaturation of the H₂O resonance, 45 degree read-pulse, over an 8000 Hz spectral width, 4 s acquisition time, centered on the water peak.

2. Page 6, "This suggests that chemical features unique to OD Cu (e.g., sub-surface oxygen) are probably absent in the structured In₂O₃/Cu₂O electrodes". I do not agree that the high activity of OD Cu came from the sub-surface oxygen. Although some semi-in situ techniques indicated there would be the presence of oxygen even at very low potentials, a recent study by Joel W. Ager (Yanwei Lum, Joel W. Ager, *Angewandte Chemie International Edition*, 2017, DOI: 10.1002/anie.201710590) suggested the previously detected sub-surface oxygen is due to the very quick oxidation of Cu when exposed to air even at a very short period. The high activity is more possibly originated from the grain boundaries.

3. Page 9, "this observation would be consistent with a higher density of grain boundaries in the irregular Cu₂O-derived layer, since the diffusion coefficient along the grain boundaries in the Cu-In system is six orders of magnitude higher than through the lattice. Nevertheless, we remark that

additional studies on the solid-state diffusion kinetics of the Cu-In system under eCO₂RR conditions would be necessary to further rationalize this behaviour." I am curious whether In₂O₃ islands can be fabricated on an oxide-derived Cu thin film (with lots of grain boundaries) to support this hypothesis? If achievable, then the solid-state diffusion can be evaluated under electrolysis/non-electrolysis conditions, and the contribution from natural and potential-induced diffusion can be deconvoluted.

NCOMMS-17-32203-T – Response to Reviewers

Comments in blue - Replies in black - Actions in bold

Reviewer #1

Authors carry out an investigation of CO₂ reduction at microfabricated electrodes, varying the composition and the In/Cu interface length to gain insights in the mechanism of formation of CO. The work is interesting and provides some insight into the mechanism, since the combination InO_x/CuO_y appears to perform better, suggesting an effect at the interface. Besides this insight however the mechanism of CO₂ reduction at Cu-In based systems does not appear to be better understood. The authors do not start with a mechanistic hypothesis to verify, but they only hypothesize an effect at the interface. This hypothesis in other works has been hypothesized and perhaps verified in terms of a bimetallic effect, which has been mentioned by the authors at the start of the manuscript. Overall, I do not think that this work provides sufficient insight for publication in this journal.

As noted by the Reviewer, several studies have reported enhanced performance toward CO evolution over copper-indium catalysts, but the origin of this synergistic effect remains poorly understood and there is still considerable uncertainty in regard to the nature of the active phase. Takanahe et al. (*Angew. Chem. Int. Ed.*, **2015**, 54, 2146) associated the enhanced performance to the formation of an alloy (i.e., Cu₁₁In₉). However, this link has not been conclusively verified. For instance, bulk Cu₁₁In₉ electrodes prepared directly from the metallic components perform poorly, and a solely bimetallic effect would not seem to explain the enhancement of the performance observed in the presence of oxidic indium phases (Pérez-Ramírez et al., *J. Catal.*, **2016**, 343, 266 and *ACS Catal.*, **2016**, 6, 6265). These latter findings led to our initial hypothesis that the synergistic effect in copper-indium catalysts could be linked to the existence of bifunctional sites at metal-oxide interfaces, rather than to the bimetallic compositions. Given this complex picture, the evaluation of this hypothesis required the preparation of catalysts with a high degree of structural and compositional control, motivating the development of the microfabrication approach herein presented.

The evaluation and characterization of the microfabricated electrodes disclosed new insights into the composition and the formation mechanism of the active phase in copper-indium catalysts. For instance, the results of this work point toward bimetallic phases that contain only low amounts of indium, rather than metal-oxide interfaces or practically equimolar alloys, as the origin of the high eCO₂RR activity of copper-indium catalysts. Furthermore, we show that the active phase is formed at the vicinity of the interfaces due to the diffusion of indium into the copper matrix, and that this process is driven by the reduction of Cu₂O in contact with an indium source upon exposure to reaction conditions. In this context, we are confident that this study contributes significantly to a better understanding of copper-indium catalysts in CO₂ reduction, and that it demonstrates the potential of the microfabrication-based approach to rationalize multicomponent catalysts.

We agree with the Reviewer that the insights from this study do not translate into an increased understanding of the mechanism of CO₂ reduction over copper-indium electrodes. However, we remark that developing a robust mechanistic understanding will depend on theoretical efforts (based on the elucidation of the actual active phase, to which this work contributes) as well as on *in situ* and *operando* spectroscopic experiments aimed at the observation of relevant adsorbates (for which the capacity of microfabrication to produce very regular microscale structures might be helpful). Ongoing studies beyond the scope of this article are expected to contribute along these directions.

Furthermore, we agree with the Reviewer that the description of the hypothesis to be tested and of its relevance to the understanding of copper-indium catalysts (particularly in regard to the elucidation of the active phase) were unclear in the original version of the manuscript. Likewise, we realized that we had been elusive in key parts of the manuscript regarding the interpretation of the results. We thank the Reviewer for bringing this issue to our attention.

The introduction of the manuscript has been thoroughly revised and the initial hypothesis has been more clearly stated. Similarly, a more complete context of the current state of the understanding of copper-indium catalysts has been added (line 40). Furthermore, the discussion has been revised to eliminate vague statements and highlight the relevance of the findings in regard to the composition and formation mechanism of the active phase in copper-indium catalysts.

The abstract is long and at various places repetitive and not very clear; some statements are cryptic.

We agree on this appreciation, as also highlighted by Reviewer #2. The revised version defines more clearly the main results toward a better understanding of Cu-In catalysts, i.e., the key role of the oxidic copper, the in situ formation of the active phase at the copper-indium interface, and its indium-poor bimetallic nature. In view of the exploratory extension to the Cu-Sn system added to the revised manuscript (**line 291 and Supplementary Fig. 10**), the abstract also highlights the versatility of microfabrication. We also made a parallel clarification effort in the Discussion, separating more notoriously advantageous general features linked to microfabrication from the concrete insights obtained upon application to the Cu-In system. As a result, **Abstract and Discussion have been extensively modified.**

The authors mention that they attempt to limit compositional changes of the catalyst by keeping the experiments short. In reality, indium oxidation may occur at very short times, and In or In₂O₃ may transform in In(OH)_x.

As stated in the manuscript, the short duration of the experiments aimed to reduce the extent of structural evolution of the electrodes so that their post-reaction characterization could be related in a more direct way to their initial state. However, fast compositional changes (e.g., reduction of the Cu₂O substrate and In₂O₃ islands) were indeed expected upon exposure to reaction conditions, which are evidenced by the chronoamperometric curves added to the revised version as **Supplementary Fig. 4** and more thoroughly discussed in **line 130 and line 179** of the revised text. Nevertheless, we remark that under the cathodic operation conditions of the eCO₂RR only reduction processes are favored. Consequently, indium oxidation would not occur under CO₂ reduction conditions but rather upon return to open circuit potential following the completion of the electrolysis, or subsequent exposure to air. In this context, ToF-SIMS and XPS analyses of post-reaction electrodes did detect trace amounts of hydroxide, which is consistent with the oxidation of indium following the reaction and with our previous work on the evolution of Cu-In catalysts (*ACS Catal.* **2016**, *6*, 6265). **The ToF-SIMS and XPS analyses and associated discussions have been added in the revised manuscript (Supplementary Figs. 8 and 9 and lines 247 and 251, respectively).**

The surface diffusion during the reduction tests are due to gradients of electrochemical potential; these should be expected and their magnitude can be very different from conventional diffusion.

We appreciate this comment. With this idea in mind, and in line with a suggestion from Reviewer #3, we have conducted an additional experiment to compare diffusion processes of metallic indium in the absence/presence of potential over an electrochemically pre-reduced Cu₂O layer (labelled as In/OD Cu in the revised manuscript). Electrodes were stored for 6 days in air and were analyzed by EDX prior and post electrolysis. Chemical maps surprisingly revealed that diffusion of indium did not happen to an observable extent in either case, thus revealing a diffusion process driven by both the electrochemical potential gradient and the restructuring of the reduced oxidic copper phase. **We have added catalytic and characterization results over In/OD Cu and discussed them in the revised manuscript (line 266 and Fig. 7).**

The attempt to determine In surface diffusion should have been performed with a XPS, not with EDAX.

We agree with the reviewer on the value of surface-sensitive techniques to study the extent of the surface diffusion of indium and help elucidate the identity of the active phase formed around the interfaces. In the case of the EDAX measurements, the accelerating voltage (5 kV) was minimized to ensure representativeness of the surface (penetration depth estimated as 100-200 nm). Furthermore, in addition to the EDAX analysis, we have now performed XPS measurements on a post-reaction $\text{In}_2\text{O}_3/\text{Cu}_2\text{O}$ electrode ($d = 50 \mu\text{m}$) by focusing the X-ray beam on three different regions: (1) the bare substrate, (2) the interfacial region containing the halo, and (3) an island containing the halo. This analysis revealed the presence of oxidized indium within the islands (in accordance with ToF-SIMS), but the limited spatial resolution of conventional XPS equipment makes it difficult to assign any spectroscopic features exclusively to the halo region, hindering the analysis of the surface diffusion of indium. This limitation is apparent when one considers that the spot size of the X-ray beam ($20 \mu\text{m}$) is much larger than the thickness of the halos observed by EDAX (ca. $1 \mu\text{m}$). Furthermore, non-negligible photoemission from outside the nominal analysis area is still recorded due to the long tail of the primary X-ray beam (Scheithauer, *Surf. Interface Anal.*, **2008**, 40, 706). These limitations complicate the use of conventional XPS instruments for spatially-resolved analyses at the characteristic length scales of the microfabricated electrodes herein studied. However, we believe that the high degree of geometrical control possible with microfabrication in combination with more specialized spectroscopic equipment with higher spatial resolution (e.g., NanoESCA, ideally with a synchrotron source) can provide even more detailed insights into the phenomena reported in this work. **The spectra from XPS analyses carried out on post-reaction $\text{In}_2\text{O}_3/\text{Cu}_2\text{O}$ electrodes (Supplementary Fig. 8) and the corresponding discussion (line 251) have been added to the revised version.**

My interpretation of these data is that Cu and Indium interdiffuse to make a better catalyst due to the bimetallic effect. Cu and In tend to interdiffuse in the solid state as shown by several works showing successive electrodeposition of In over Cu, forming an alloy in a short time. As such, the work of Hoffman (ref. 22) shows a better way to maximize (but not control) the interface length between Cu and In.

We agree with the general picture described by the Reviewer regarding the diffusion of indium in the copper matrix. In this concrete aspect, microstructured electrodes added precise information about the relevance of the copper source and the required interplay with the electrochemical environment, thus revealing the real complexity inherent to this system. We note that maximization of the interface length was not an absolute target in our study, rather achieving a range of interfacial densities wide enough to unambiguously identify activity correlations.

Reviewer #2

The manuscript provides insight into microfabrication techniques for fabrication of controlled electrodes in order to precisely evaluate the role of metal/metal-oxide catalysts on CO₂ reduction. The experiments are well designed and novel, and the results would be of interest for wide range of experts in electrocatalysis and microfabrication. The manuscript can be published with minor revision, after considering below comments:

We thank Reviewer #2 for the thorough reading and for recognizing the significance of the findings.

1. "Abstract" and "conclusions" are a bit vague, please restructure and mention the core point and findings of your results.

This aspect was also stressed by Reviewer #1. Accordingly, **we have rewritten both parts in the revised version with emphasis in highlighting the wide scope of this approach for catalytic**

studies and clearly enumerating the main conclusions upon its application to the Cu-In system shedding light on the nature and formation process of the bimetallic active phase.

2. The whole manuscript needs an English proof read. there are many long sentences, hard to follow and understand, specially in:

page 4 line 4; page 5 line 5-9; page 6 line 14-18, page 7 line 10-14; page 9 line 21-23; page 10 line 14-17; page 11 line 10-17.

We have meticulously revised the text and rephrased some paragraphs with care on improving conciseness and clarity, particularly in the lines indicated by the Reviewer.

3. Some Figures can be reorganized from supporting info to the main text to enhance reading and understanding. For example supp. 1) Fig. 6 can be also added in Fig. 2.

Caption of Supp. Fig 1 is confusing. Supp. I suggest that Figure 1a can be moved to the main text to figure 1 to help understanding the microfabricated fresh electrode and their size difference. Figure 3b can be moved to the Supp. The description for Info. Supp. Fig. 5 is not clear well.

Thank you for the various suggestions. **We have reorganized most of the figures and revised the captions to make sure the key messages are properly conveyed.**

4. Supp. Figure 2: 1) correct the caption (-:6 V) 2) Please note the In₂O₃ peaks. 3) It would be also useful to add XRD of In/CuO₂ or In₂O₃/Cu (used) to investigate whether any phase transition occurred.

We must note that the nanometric thickness of the islands together with their mostly amorphous nature did not allow detection of In-related peaks in the XRD patterns in most cases (fresh or used), thus invalidating it as a tool to monitor phase transitions. **The caption has been revised and the position of main peaks associated to In₂O₃ and In, and the diffractogram of In₂O₃/Cu (used) have been added for the sake of completeness (Supplementary Fig. 2).**

Reviewer #3

This work by Javier Pérez-Ramírez et al., a precisely controlled interface density was created by the microfabrication technique, which provides an excellent platform to analyze the impacts of these interfaces. Suitable control experiments were relatively well conducted in combination with advanced characterization techniques. The proposed solid-state diffusion mechanism add new insight into the origin of activity on this bimetallic catalysts, and this strategy is promising in understanding the catalytic behaviors of other multicomponent catalysts. Overall, this manuscript was well written with persuasive results, and is potentially suitable for publication. However, a few minor issues need to be addressed before acceptance can be suggested.

We gratefully thank Reviewer #3 for the positive feedback and suggested improvements. We have revised his/her comments and developed related experimental work to strengthen the manuscript. Additionally, **we have included new surface characterization by ToF-SIMS and XPS (line 247 of the revised manuscript and Supplementary Figs. 8 and 9) and new experiments extending the microfabrication process to the SnO₂/Cu₂O system, in order to compare the in situ modification of copper by Sn- and In-based materials (line 291 and Supplementary Fig. 10), which additionally shows the versatility associated to microfabrication.**

1. "A short reaction time (5 min) was selected to reduce the interference of bubbles that gradually accumulate on the electrode surface and to limit the extent of restructuring, so that any changes observed in the electrodes could be related to their initial structure while still allowing the assessment of the catalytic activity." For most catalysts, the current would drop fast at the initial stage of electrolysis, and then reach semi-steady state after a few minutes, could the author show the current-

time (i-t) curves? If the current decays quickly, the faradaic efficiency obtained at the beginning may not be accurate due to the head-space equilibrium issue. However, I realized a very small cell was used in this study which may have a small head space as well, possibly this issue can be alleviated.

We acknowledge the importance of considering the transient state within a short reaction time of 5 min. The reaction time was selected after performing longer electrolyses and assuring that results were fully representative of the steady state. Current densities reported in our study are the final values obtained at the end of each 5-min electrolysis and typically represent 90-100% of the steady one. In parallel, and as mentioned by the Reviewer, the small headspace volume of the cathodic compartment (ca. 2 cm³) allowed its quick equilibration, leading to reproducible quantification results, though this effect might still show some influence in underestimating current efficiencies. Fortunately, the observed variations in activity across different materials and geometries span several orders of magnitude, thus discarding inaccuracies due to the initial current decay as relevant for the conclusions. **In the revised version we have included a comparison of results between 5- and 20-min electrolyses for a set of In₂O₃/Cu₂O electrodes evidencing the representativeness of the selected reaction time (line 134 and Supplementary Fig. 4).**

In addition, since the total faradaic efficiency is only ~70%, can any liquid product be detected via H NMR? Formate is a possible liquid product in In-Cu systems. The estimated concentration is over 100 μM if assuming the remaining product is formate, which maybe below the detection limit of HPLC and IC, but is sufficient in NMR analysis. 128 scans with water suppression on a 500 MHz NMR are enough to get very good signal-to-noise ratio. The following parameters can be used: 0.5 s prescan delay, 5 s presaturation of the H₂O resonance, 45 degree read-pulse, over an 8000 Hz spectral width, 4 s acquisition time, centered on the water peak.

We appreciate this comment. As indicated by the Reviewer, the concentration of liquid products in the electrolyte did not allow detection by HPLC analysis. Upon application of Reviewer's suggested parameters to analysis by NMR, we successfully quantified the selectivity to formate by analyzing both the cathodic and anodic compartments. Results indicate stable Faradaic efficiencies of 4-7% irrespective of the interfacial density, evidencing the contribution to the initial stages of the electrolysis of the reduction of the Cu and/or In phases. **The revised version includes a representative case (In₂O₃/Cu₂O) showing selectivity toward gaseous and liquid products where this effect is clearly observed (line 167 and Supplementary Fig. 6).**

2. Page 6, "This suggests that chemical features unique to OD Cu (e.g., sub-surface oxygen) are probably absent in the structured In₂O₃/Cu₂O electrodes". I do not agree that the high activity of OD Cu came from the sub-surface oxygen. Although some semi-in situ techniques indicated there would be the presence of oxygen even at very low potentials, a recent study by Joel W. Ager (Yanwei Lum, Joel W. Ager, *Angewandte Chemie International Edition*, 2017, DOI: 10.1002/anie.201710590) suggested the previously detected sub-surface oxygen is due to the very quick oxidation of Cu when exposed to air even at a very short period. The high activity is more possibly originated from the grain boundaries.

We agree with the Reviewer. During the final stage of the preparation and during the revision process of the original version of the manuscript relevant studies supporting the catalytic role of grain boundaries in the eCO₂RR have appeared, such as the mentioned study by Lum and Ager, and the one conducted by Kanan et al. (*Science*, 2017, 358, 1187). **In view of this, we have modified the corresponding discussion in the text accordingly (line 163).**

3. Page 9, "this observation would be consistent with a higher density of grain boundaries in the irregular Cu₂O-derived layer, since the diffusion coefficient along the grain boundaries in the Cu-In system is six orders of magnitude higher than through the lattice. Nevertheless, we remark that additional studies on the solid-state diffusion kinetics of the Cu-In system under eCO₂RR conditions

would be necessary to further rationalize this behaviour.” I am curious whether In_2O_3 islands can be fabricated on an oxide-derived Cu thin film (with lots of grain boundaries) to support this hypothesis? If achievable, then the solid-state diffusion can be evaluated under electrolysis/non-electrolysis conditions, and the contribution from natural and potential-induced diffusion can be deconvoluted. We fully agree with the reviewer on the relevance of the proposed experiment and thank him/her for this suggestion, since its realization added relevant insights to the manuscript. We introduced electroreduction under eCO_2RR conditions before applying the photolithographic process to Cu_2O -covered substrates, allowing the preparation of In islands over a pre-reduced Cu_2O (OD Cu) layer (named as In/OD Cu in the revised version). Surprisingly, neither the electrodes stored under ambient conditions nor the ones that, in addition, underwent electrolysis showed modification of the adjacent OD Cu substrate at the microscale (i.e., halos around the islands were absent). In close agreement, In/OD Cu electrodes exhibited similar performance to $\text{In}_2\text{O}_3/\text{Cu}$, where halos did not develop, either. We must then conclude that the reduction of the Cu_2O phase must occur in the presence of the indium phase so that the positive modification of the copper substrate takes place and that the expected larger amount of grain boundaries in the OD Cu cannot account solely for the observed results. **We have added catalytic and characterization results over In/OD Cu and discussed them in the revised manuscript (line 266 and Fig. 7)**

REVIEWERS' COMMENTS:

Reviewer #1 (Remarks to the Author):

My evaluation of this work does not change from the previous version.

Reviewer #2 (Remarks to the Author):

The comments are carefully considered and the manuscript is thoroughly modified. It can be accepted for publication.

Reviewer #3 (Remarks to the Author):

The authors addressed my concerns fully.

NCOMMS-17-32203A – Response to Reviewers. Revised version.

Comments in blue - Replies in black - Actions in bold

Reviewer #1

My evaluation of this work does not change from the previous version.

We are sorry that the improvements included in the revised version did not change the judgment of the Reviewer about the manuscript. In any case, we greatly appreciate the Reviewer's comments, which helped us sharpen its content and scope.

Reviewer #2

The comments are carefully considered and the manuscript is thoroughly modified. It can be accepted for publication.

We appreciate the positive feedback on the revised version and thank again the Reviewer for the attentive reading and constructive comments.

Reviewer #3

The authors addressed my concerns fully.

We are certainly glad the Reviewer found the added experimental data satisfactory. We must acknowledge the tangible improvement introduced in the revised manuscript as a consequence of suggestions raised by the Reviewer.